# Mineral Acid Co-Extraction in Reactive Extraction of Lactic Acid Using a Thymol-Menthol Deep Eutectic Solvent as a Green Modifier

**DOI:** 10.3390/molecules29081722

**Published:** 2024-04-11

**Authors:** Paul Demmelmayer, Marija Ćosić, Marlene Kienberger

**Affiliations:** Institute of Chemical Engineering and Environmental Technology, Graz University of Technology, Inffeldgasse 25/C, 8010 Graz, Austria; demmelmayer@tugraz.at (P.D.); cosic.m@outlook.com (M.Ć.)

**Keywords:** natural deep eutectic solvent, reactive extraction, lactic acid extraction, mineral acid extraction

## Abstract

Carboxylic acids can be isolated from fermentation broths using reactive liquid-liquid extraction, offering an alternative to the environmentally harmful state-of-the-art process of precipitating calcium lactate. To enhance the sustainability of liquid-liquid extraction processes, greener solvents, such as natural deep eutectic solvents, are investigated. However, fermentation broths often exhibit pH values unsuitable for carboxylic acid extraction, which can be adjusted using mineral acids, though mineral acids may be co-extracted. In this study, we systematically examine the co-extraction of hydrochloric, nitric, sulfuric, and phosphoric acid during extraction and back-extraction of lactic acid. The solvent phase consisted of tri-*n*-octylamine, trioctylphosphine oxide, or tributyl phosphate diluted in a thymol-menthol deep eutectic solvent. The back-extraction was conducted using a diluent swing with *p*-cymene as the antisolvent and water as the receiving phase. Tri-*n*-octylamine showed the highest efficiency for lactic acid (up to 29.8%) but also the highest co-extraction of mineral acids (up to 50.9%). In contrast, trioctylphosphine oxide exhibited a lower but more selective lactic acid extraction (5.94%) with low mineral acids co-extraction (0.135%). Overall, the highest co-extraction was observed for phosphoric acid and the lowest for nitric acid. In conclusion, the selected solvent phase composition and mineral acid influence the co-extraction and, thus, final product purity. The successful application of the natural deep eutectic solvent as the modifier enhances the sustainability of liquid–liquid extraction processes.

## 1. Introduction

Combating environmental pollution and the associated climate change is a present global challenge. To mitigate this crisis, there is growing emphasis on the transition from a fossil-based industry to a bio-based industry. Biorefineries produce valuable chemicals (e.g., carboxylic acids or phenols), fuels, and other materials from sustainable resources, offering a green alternative to fossil-based refineries.

Lactic acid (LA), or 2-hydroxypropanoic acid according to IUPAC, is a bio-based bulk chemical of great interest because of its various applications, e.g., chemical industry, food industry, cosmetics, or monomer for the production of poly-lactic acid [1]. The global LA market was valued at USD 67.9 billion in 2023, with an expected compound annual growth rate of 10.4% until 2032 [2]. About 90% of the global LA production is carried out via fermentation using microorganisms, such as yeast or bacteria. One advantage of the fermentative production pathway is to yield optically pure L- or D-LA; in contrast, the chemical synthesis of LA leads to a racemic mixture [3]. A second advantage of fermentative LA production is that renewable resources or residues, such as corn stover [4], wheat straw [5], or sweet sorghum [6], can be used. The main cost-driving factor in LA production is downstream processing accounting for up to 60% of the overall production costs [7,8]. The state-of-the-art LA downstream processing involves the precipitation of calcium lactate from the fermentation broth using a calcium base and then converting the calcium lactate to LA through acidification with sulfuric acid. In addition to the high chemical demand, this process generates one mole of gypsum sludge per mole of produced LA, resulting in environmental issues [9]. Therefore, alternative isolation methods for LA, such as reactive liquid-liquid extraction (LLE), are investigated [8,10].

In reactive LLE, LA is transferred from the aqueous fermentation broth into an immiscible, organic solvent phase via a reversible reaction between LA and the reactive extractant in the solvent phase. Commonly applied solvent phases for LA extraction consist of a reactive extractant (e.g., tri-*n*-octylamine (TOA) or trioctylphosphine oxide (TOPO)), a modifier (e.g., octanol or decanol), and/or a diluent (e.g., octane or undecane) [11]. The modifier increases the solubility of the LA-extractant complex, whereas the diluent influences the properties of the solvent phase, such as density or viscosity [10,12]. Reactive extractants are categorized according to their extraction mechanisms, such as anion exchangers (e.g., secondary or tertiary amines) or solvating agents (e.g., phosphine oxides or alkyl-substituted phosphates) [13]. Besides its advantages, such as large production capacity, low energy consumption, and continuous operation, reactive LLE has the drawback of mainly applying fossil-based chemicals [14]. To enhance the sustainability of the process, alternative solvents, such as natural deep eutectic solvents (NADES), are investigated. NADES are eutectic mixtures consisting of two or more naturally occurring compounds. These mixtures show a negative deviation from the ideal eutectic behavior and, thus, exhibit an even stronger melting point depression compared to ideal eutectic mixtures. The melting point depression results from intermolecular interactions, such as hydrogen bonding or van der Waals interactions [15,16]. As NADES consist of naturally derived compounds, they are considered biocompatible, environmentally friendly, and non-toxic [17]. Nevertheless, toxicological studies are still scarce, and the available literature only investigates the toxicity of hydrophilic NADES. However, the available studies show that the toxicity of hydrophilic NADES depends on their constituents and concentration [18,19,20,21,22]. The variety of possible NADES compounds enables the production of task-specific NADES, which can substitute fossil-based solvents in LLE processes to promote sustainable practices within the chemical processing industry [23].

Depending on the applied reactive extractant, either the protonated or deprotonated form of the LA molecule is preferably extracted. For example, the tertiary amine TOA mainly extracts protonated LA molecules by ion-pair formation [6]. This emphasizes the importance of the pH value of the aqueous feed phase together with the p*K*_a_ value of LA (see Figure 1).

Industrial LA fermentations are usually carried out at pH values between five and seven, which is well above the p*K*_a_ value of LA. Thus, the LA molecules are present in their deprotonated form. One exception is the fermentation technology implemented by Cargill in 2008, which uses a genetically modified yeast capable of producing LA at pH values below three [25]. Besides more advanced methods of pH adjustment, such as electrochemical pH swing [26], a common method to adjust the pH value of aqueous solutions is the addition of mineral acids, such as sulfuric acid (H_2_SO_4_) [27,28]. Additionally, mineral acids can be found in the aqueous phase, serving as catalysts, for example, in the production of levulinic acid, where H_2_SO_4_ is used for the hydrolysis of lignocellulosic materials [29]. Mineral acids are co-extracted to some extent, diminishing the available extractant molecules for the extraction of LA [30].

The investigation of mineral acid co-extraction has been conducted for different reactive LLE systems with fossil-based modifiers and diluents [29,30,31]. However, an open research question is the assessment of mineral acid co-extraction when NADES are applied in reactive LLE. Additionally, research efforts must extend beyond the extraction step to assess the mineral acid co-back-extraction during solvent phase regeneration.

To address this research gap, we investigate the co-extraction of the four mineral acids hydrochloric acid (HCl), nitric acid (HNO_3_), H_2_SO_4_, and phosphoric acid (H_3_PO_4_) during reactive LA extraction and back-extraction. As reactive extractants, TOA, TOPO, and tributyl phosphate (TBP) were used, which are commonly applied for carboxylic acid extraction [32,33,34]. The extractants were diluted in a thymol-menthol-based NADES consisting of 60 mol% thymol and 40 mol% menthol (tmDES is further used as a shortcut for this specific NADES). As a benchmark, we employed the frequently used modifier 1-octanol and the green diluent limonene. The back-extraction step was realized by a diluent swing where the extract phase was mixed with an antisolvent, and water was provided as the receiving phase. The antisolvent, a nonpolar compound (e.g., heptane [35]), is not able to accept or donate hydrogen bonds and, thus, reduces the distribution ratio of LA in the solvent phase [36]. As a result, the LA is transferred into the receiving phase. In the present study, we used *p*-cymene as an antisolvent because it can be produced from renewable resources, e.g., it is a by-product of sulfite pulping [37]. Performing the back-extraction at an elevated temperature of 70 °C further lowers the distribution ratio of LA [34,38,39]. This approach has the advantage, as compared to back-extraction with sodium hydroxide solution [40,41], that the acid is obtained in its protonated form and not in its salt form. Thus, a hydrolysis step to obtain the acid form is not required. The present work provides a systematical approach for the co-extraction of the mineral acids HCl, H_2_SO_4_, HNO_3_, and H3PO_4_ during both LA extraction and back-extraction, utilizing reactive extractants mixed with a NADES. Substituting the modifier and/or diluent, which account for the major part of the solvent phase, with a NADES enhances the sustainability of the LLE process.

## 2. Results and Discussion

### 2.1. Physical Extraction of Mineral Acids

The physical extraction of HCl, HNO_3_, H_2_SO_4_, and H_3_PO_4_ by 1-octanol, tmDES, and limonene was investigated using single-acid model solutions containing one of the mentioned mineral acids. The starting concentrations of the single-acid model solutions were 0.449 ± 0.004, 0.438 ± 0.004, 0.443 ± 0.004, and 0.458 ± 0.004 mol·L^−1^ for HCl, HNO_3_, H_2_SO_4_, and H_3_PO_4_, respectively. Table 1 summarizes the initial feed pH value (pH_F_), raffinate pH value (pH_R_), and mole-based distribution ratio of the mineral acids for the extraction step (*D*_extr,mineral_).

The results in Table 1 show that all solvent phases exhibit a low mineral acid extraction with the *D*_extr,mineral_ ranging from 0.0359 ± 0.0042 to 0.0877 ± 0.0032. Brouwer et al. [29] reported negligible sulfuric acid extraction for various solvents (e.g., 1-hexanol, 1-octanol, or dodecane) due to the unfavorable solvation of deprotonated acid molecules in the hydrophobic solvents. Despite the presence of hydroxyl groups in 1-octanol and tmDES, no higher *D*_extr,mineral_ is observed for these two modifiers as compared to limonene. As a result of the low extraction of mineral acids, the pH value only marginally changes in Table 1.

### 2.2. Reactive Extraction of Mineral Acids from Single-Acid Model Solutions

To examine the reactive extraction of HCl, HNO_3_, H_2_SO_4_, and H_3_PO_4_ from the single-acid model solutions, the reactive extractants TOA, TOPO, and TBP diluted in 1-octanol, tmDES, or limonene were applied. The single-acid model solutions used as the feed phases had the same initial acid concentrations and pH_F_ as in Section 2.1. Table 2 summarizes the results for pH_R_ and *D*_extr,mineral_. The results for extractant loading (*z*_acid_) can be found in Appendix A.

Tertiary amines, such as TOA, extract mineral acids via ion pair formation [29]. Furthermore, protonated amine molecules are known to have a higher affinity towards anions of strong acids compared to anions of weak acids [42,43]. Therefore, the highest *D*_extr,mineral_ for TOA in Table 2 are attained for HCl and the lowest *D*_extr,mineral_ for H_3_PO_4_. However, differences between 1-octanol, tmDES, or limonene are observed. The solvent phase TOA:1-octanol shows the highest *D*_extr,mineral_ with values of 22.5 ± 0.1, 22.3 ± 0.2, 17.3 ± 0.1, and 16.1 ± 0.0 for HCl, HNO_3_, H_2_SO_4_, and H_3_PO_4_, respectively. Besides the polar hydroxyl group in 1-octanol, van der Waals forces between the linear carbon chain of 1-octanol and the octyl chains of TOA result in further stabilization of the acid–amine complex in the extract phase. With tmDES and limonene, a decrease in *D*_extr,mineral_ is observed with values of 1.27 ± 0.04 and 1.18 ± 0.01 for HCl, 1.27 ± 0.03 and 1.23 ± 0.01 for HNO_3_, 0.776 ± 0.008 and 0.686 ± 0.007 for H_2_SO_4_, and 0.402 ± 0.009 and 0.386 ± 0.006 for H_3_PO_4_, respectively. Interestingly, limonene only shows slightly lower distribution ratios as compared to tmDES although it does not contain hydroxyl groups to stabilize the acid–amine complex. One explanation is that the unsaturated double bonds in the limonene structure contribute to the extraction of mineral acids or improve the stabilization of the acid–amine complex. More detailed investigations regarding the extraction mechanism are required in the future.

In contrast to TOA, the protonation of the two phosphorous-based extractants TOPO [44] and TBP [45,46] is only possible at a high proton activity. The p*K*_a_ value of TBPH^+^ is reported to be below one, e.g., Lommelen and Binnemans [46] used a p*K*_a_ of −0.5 for TBPH^+^ in their simulations. With the starting pH value of 0.46 ± 0.01 to 1.15 ± 0.01 of the single-acid model solutions, the protonation of TOPO and TBP can be neglected. Consequently, the extraction of mineral acids by these two extractants mainly takes place via hydrogen bonding of the protonated mineral acid molecules to the oxygen atom at the phosphorus atom [29,46,47]. This explains the low *D*_extr,mineral_ in the case of HCl, HNO_3_, and H_2_SO_4_ as these acids are mainly present in their deprotonated form at the investigated pH values. When using TOPO or TBP as an extractant, the *D*_extr,mineral_ ranges from 0.0100 ± 0.0018 to 0.0746 ± 0.0029 for HCl, from 0.0239 ± 0.0069 to 0.0853 ± 0.0010 in the case of HNO_3_, and from 0.0332 ± 0.0049 to 0.0888 ± 0.0016 for H_2_SO_4_. Slightly higher *D*_extr,mineral_ are obtained for H_3_PO_4_ (0.114 ± 0.015 to 0.130 ± 0.011), which can be attributed to its p*K*_a_ value of the first dissociation step of 2.16 [48] and the starting pH value of the H_3_PO_4_ single-acid model solution of 1.15 ± 0.01. Based on the dissociation of H_3_PO_4_, about 90% of the acid molecules are protonated and, hence, can be extracted by TOPO and TBP.

### 2.3. Reactive Extraction of Mineral Acids from a Multi-Acid Model Solution

To investigate the effect of the presence of multiple mineral acids in the feed solution on the distribution coefficient, the same solvent phases as in Section 2.2 were used with a model solution containing the four mineral acids HCl, HNO_3_, H_2_SO_4_, and H_3_PO_4_. Initial acid concentrations were 0.101 ± 0.001, 0.103 ± 0.001, 0.104 ± 0.001, and 0.105 ± 0.001 mol·L^−1^ for HCl, HNO_3_, H_2_SO_4_, and H_3_PO_4_, respectively. The pH_F_ of the multi-acid model solution was 0.597 ± 0.006. Results for the pH_R_, total extractant loading (*z*_tot_), and *D*_extr,mineral_ are summarized in Table 3. The results for *z*_acid_ are summarized in Appendix A.

The comparison of the *D*_extr,mineral_ of the four mineral acids when using TOA as an extractant reveals that the highest *D*_extr,mineral_ are obtained for the strongest acids HCl and HNO_3_, followed by H_2_SO_4_. H_3_PO_4_, the weakest acid, exhibits the lowest *D*_extr,mineral_ when using TOA as reactive extractant. Moreover, *D*_extr,mineral_ decreases from 1-octanol to tmDES to limonene. Using the solvent phase TOA:1-octanol, *D*_extr,mineral_ is 2.06 ± 0.01, 3.26 ± 0.03, 1.09 ± 0.01, and 0.176 ± 0.002 for HCl, HNO_3_, H_2_SO_4_, and H_3_PO_4_, respectively, at a *z*_tot_ of 1.07 ± 0.02. In contrast, the *z*_acid_ in Appendix A, which corresponds to *z*_tot_ as only one acid is present in the feed phase, for TOA:1-octanol with the single-acid model solutions in Section 2.2 is 2.07 ± 0.00, 2.01 ± 0.00, 2.02 ± 0.00, and 2.09 ± 0.00 for HCl, HNO_3_, H_2_SO_4_, and H_3_PO_4_, respectively. This shows that the presence of multiple acids in the feed phase lowers the extractant loading. When diluting TOA in the tmDES or limonene, the *D*_extr,mineral_ is 1.82 ± 0.02 and 1.19 ± 0.01 for HCl, 1.87 ± 0.02 and 1.28 ± 0.01 for HNO_3_, 0.573 ± 0.010 and 0.392 ± 0.002 for H_2_SO_4_, and 0.171 ± 0.002 and 0.150 ± 0.019 for H_3_PO_4_, respectively. The *z*_tot_ of 0.933 ± 0.002 for TOA:tmDES and 0.783 ± 0.008 for TOA:limonene shows that also for tmDES and limonene, lower extractant loadings are obtained for the multi-acid solution as compared to the single-acid solutions; *z*_acid_ for the single-acid solutions in Appendix A ranges from 0.624 ± 0.008 to 1.31 ± 0.01.

The phosphorous-based extractants TOPO and TBP diluted in 1-octanol, tmDES, or limonene show similar *D*_extr,mineral_ for all four mineral acids with the *D*_extr,mineral_ ranging from 0.0118 ± 0.0037 to 0.0497 ± 0.0019 for HCl, from 0.0209 ± 0.0066 to 0.0551 ± 0.0067 in the case of HNO_3_, from 0.0231 ± 0.0063 to 0.0487 ± 0.0056 for H_2_SO_4_, and from 0.0154 ± 0.0053 to 0.0417 ± 0.0055 for H_3_PO_4_. As a result of the low *D*_extr,mineral_, TOPO and TBP exhibit a low *z*_tot_ with values of 0.0486 ± 0.0062 to 0.0840 ± 0.0096.

In conclusion, Section 2.2 and Section 2.3 show that the tertiary amine TOA shows the highest extraction of the mineral acids HCl, HNO_3_, H_2_SO_4_, and H_3_PO_4_. However, TOA diluted in 1-octanol, tmDES, or limonene results in different *D*_extr,mineral_ with 1-octanol showing higher *D*_extr,mineral_ as compared to tmDES and limonene. In contrast to TOA, the extractants TOPO and TBP show a low mineral acid extraction for 1-octanol, tmDES, and limonene.

### 2.4. Selectivity of Reactive Lactic Acid Extraction in the Presence of Mineral Acids

To adjust the feed pH value in carboxylic acid extraction, mineral acids, such as H_2_SO_4_ [6,49], are commonly applied. However, as shown in Section 2.2 and Section 2.3, especially strong mineral acids are extracted by tertiary amines. This suggests that mineral acid molecules used for pH adjustment in carboxylic acid extraction are co-extracted and, thus, reduce the number of extractant molecules in the solvent phase available for carboxylic acid extraction. As a result, the carboxylic acid extraction is limited, while the optimum pH value for carboxylic acid extraction prevails. To quantify the co-extraction of the mineral acids HCl, HNO_3_, H_2_SO_4_, and H_3_PO_4_ during reactive extraction of LA, the extractant TOA diluted in 1-octanol, tmDES, or limonene was applied. In addition, the solvent phase TOPO:tmDES was investigated. The starting concentrations of the model solutions were 0.196 ± 0.003, 0.209 ± 0.002, 0.206 ± 0.002, 0.201 ± 0.002, and 0.203 ± 0.002 mol·L^−1^ for LA, HCl, HNO_3_, H_2_SO_4_, and H_3_PO_4_, respectively. Table 4 summarizes the pH_F_, pH_R_, *z*_tot_, mole-based distribution ratio of LA for the extraction step (*D*_extr,LA_), *D*_extr,mineral_, and selectivity of the LA extraction (*S*_extr,LA_) for the conducted experiments. The results for *z*_acid_ can be found in the Appendix A.

For all four mineral acids, the solvent phase TOA:1-octanol exhibits the highest LA extraction. The values for *D*_extr,LA_ decrease in the order H_3_PO_4_ > HCl > H_2_SO_4_ > HNO_3_. The solvent phase TOA:tmDES exhibits a similar trend as the *D*_extr,LA_ decreases in the order H_3_PO_4_ > HCl > HNO_3_ > H_2_SO_4_. In terms of mineral acid co-extraction, TOA:1-octanol and TOA:tmDES show similar results. The highest *D*_extr,mineral_ is observed for HNO_3_ with values of 21.5 ± 0.3 and 13.0 ± 0.1 for TOA:1-octanol and TOA:tmDES, respectively. For the other mineral acids, *D*_extr,mineral_ is 18.9 ± 0.2 and 7.42 ± 0.37 for HCl, 1.56 ± 0.02 and 1.27 ± 0.01 for H_2_SO_4_, and 1.18 ± 0.04 and 0.836 ± 0.008 for H_3_PO_4_ when using TOA:1-octanol and TOA:tmDES, respectively. The high *D*_extr,LA_ when using H_3_PO_4_ for pH adjustment can be explained by the low *D*_extr,mineral_ of H_3_PO_4_. As a result, fewer TOA molecules are occupied by H_3_PO_4_ molecules and, thus, are available for LA extraction. The second highest *D*_extr,LA_ is obtained with HCl, although HCl exhibits the second highest *D*_extr,mineral_. This might be explained by the chlorine anion being the smallest anion of the four mineral acids. As a result, the TOA molecule is sterically easier to access by an LA molecule as in the case of the other mineral acids. This makes the overloading of a TOA molecule more likely. Overloading is the extraction of more than one acid molecule per extractant molecule, indicated by an extractant loading higher than one. Interestingly, no clear difference in *D*_extr,LA_ between HNO_3_ and H_2_SO_4_ can be observed for TOA:1-octanol and TOA:tmDES regardless of the higher *D*_extr,mineral_ of HNO_3_ as compared to H_2_SO_4_. The solvent phase TOA:limonene results in similar *D*_extr,LA_ for H_2_SO_4_ (0.145 ± 0.002), H_3_PO_4_ (0.128 ± 0.012), HCl (0.116 ± 0.003), and HNO_3_ (0.106 ± 0.006). For the mineral acids, TOA:limonene shows the same trend as the solvent phases TOA:1-octanol and TOA:tmDES with *D*_extr,mineral_ decreasing in the order HNO_3_ > HCl > H_2_SO_4_ > H_3_PO_4_.

In contrast to the solvent phases containing TOA as the reactive extractant, TOPO:tmDES shows low *D*_extr,LA_ (0.0728 ± 0.0036 to 0.0828 ± 0.0023) and low *D*_extr,mineral_ (0.0207 ± 0.0004 to 0.0261 ± 0.0002). Brouwer et al. [29] explained the low co-extraction of H_2_SO_4_ with TOPO by the fact that TOPO is a solvating extractant as compared to TOA which is an anion-active extractant. The low *D*_extr,LA_, in combination with the even lower values for *D*_extr,mineral_, results in higher *S*_extr,LA_ for TOPO:tmDES as compared to the solvent phases containing TOA. *S*_extr,LA_ with TOPO:tmDES is 3.64 ± 0.04 for HNO_3_, 3.56 ± 0.03 for HCl, 3.29 ± 0.03 for H_3_PO_4_, and 2.63 ± 0.03 for H_2_SO_4_. For solvent phases containing TOA, the highest *S*_extr,LA_ are achieved with H_3_PO_4_ (0.527 ± 0.005 to 0.948 ± 0.009), followed by H_2_SO_4_ (0.122 ± 0.001 to 0.181 ± 0.002), HCl (0.0201 ± 0.0002 to 0.0295 ± 0.0003), and HNO_3_ (0.0113 ± 0.0001 to 0.0126 ± 0.0001).

The results of this section show that the solvent phase TOPO:tmDES shows a higher selectivity towards LA as compared to solvent phases containing TOA as the reactive extractant but suffer from low *D*_extr,LA_. In contrast, solvent phases with TOA exhibit higher *D*_extr,LA_ as compared to TOPO:tmDES but result in high mineral acid co-extraction.

### 2.5. Back-Extraction of Lactic Acid

The final section of this study examines the back-extraction of LA from the extract phase. This also includes the co-back-extraction of the mineral acids. For the back-extraction experiments, *p*-cymene was applied as an antisolvent, and distilled water was used for the receiving phase. The experiments were conducted using the extract phases from the extraction experiments in Section 2.4; the acid concentrations in the extract phase are provided in the Appendix A. Table 5 summarizes the pH value of the loaded receiving phase (pH_LR_), mole-based distribution ratio of LA for the back-extraction step (*D*_back,LA_), mole-based distribution ratio of the mineral acids for the back-extraction step (*D*_back,mineral_), and selectivity of the LA back-extraction (*S*_back,LA_).

The *D*_back,LA_ in Table 5 ranges from 1.10 ± 0.04 to 4.71 ± 0.14 for the different solvent phases and mineral acids. The comparison of the *D*_back,LA_ obtained with TOA:1-octanol (1.77 ± 0.04 to 4.71 ± 0.14) and TOA:tmDES (1.10 ± 0.04 to 3.07 ± 0.11) reveals that *D*_back,LA_ is always higher for TOA:1-octanol. This might be explained by tmDES having stronger interactions, such as hydrogen bonding, with LA molecules as compared to 1-octanol, which reduces the back-extractability of LA. The high *D*_back,LA_ (2.49 ± 0.20 to 3.79 ± 0.18) obtained with the solvent phase TOA:limonene can be attributed to the hydrophobic nature of limonene leading to a lower stabilization of the acid–amine complex in the extract phase. Thereby, the release of LA into the receiving phase is facilitated. The solvent phase TOPO:tmDES shows good LA back-extraction with a *D*_back,LA_ of 2.22 ± 0.15 to 2.68 ± 0.01. However, in this case, the low *D*_extr,LA_ of 0.0728 ± 0.0036 to 0.0828 ± 0.0023 in Table 4 must be considered resulting in low LA concentrations in the extract phases.

*D*_back,mineral_ for the different solvent phases in Table 5 varies between 0.0263 ± 0.0034 and 53.1 ± 0.5. The lowest *D*_back,mineral_ are achieved for the strongest mineral acids HCl (0.0597 ± 0.0039 to 0.668 ± 0.026) and HNO_3_ (0.0463 ± 0.0019 to 0.451 ± 0.003) followed by H_2_SO_4_ (0.0263 ± 0.0034 to 0.896 ± 0.015). In contrast, the weakest acid, OA, exhibits higher *D*_back,mineral_ of 0.0624 ± 0.0040 to 53.1 ± 0.5. This leads to the conclusion that the back-extractability of mineral acids increases with decreasing acid strength. Upon comparing *D*_back,mineral_ for the different solvent phases, the solvent phase TOPO:tmDES shows the lowest *D*_back,mineral_ (0.0263 ± 0.0034 to 0.0624 ± 0.0040), followed by TOA:tmDES (0.0846 ± 0.0010 to 4.07 ± 0.07) and TOA:1-octanol (0.131 ± 0.003 to 15.5 ± 0.4). The solvent phase TOA:limonene shows a higher back-extractability of the mineral acids with a *D*_back,mineral_ ranging from 0.451 ± 0.003 to 53.1 ± 0.5. High variations in *D*_back,LA_ and *D*_back,mineral_ among the solvent phases lead to considerable differences in the *S*_back,LA_, which ranges from 0.0715 ± 0.0026 to 102 ± 10. The lowest *S*_back,LA_ (0.0715 ± 0.0026 to 5.53 ± 0.48) is attained with TOA:limonene due to its high *D*_back,mineral_, whereas the solvent phases TOA:1-octanol and TOA:tmDES exhibit a *S*_back,LA_ of 0.114 ± 0.001 to 35.9 ± 0.3. The highest *S*_back,LA_ (37.6 ± 4.9 to 102 ± 10) is observed with the solvent phase TOPO:tmDES, however, at the same time TOPO:tmDES shows the lowest *D*_extr,LA_ (Table 4).

The *D*_back,mineral_ and *D*_back,LA_ indicate good back-extractability of the mineral acids and LA from the extract phases suggesting good solvent phase recyclability. However, a more detailed investigation of the recyclability of the solvent phase is required in future work.

For a clearer assessment of the extraction and back-extraction of LA and mineral acids, Figure 2 shows the total efficiency for LA (*E*_tot,LA_) and the mineral acids (*E*_tot,mineral_), which measures the percentages of moles extracted from the feed phase in the extraction step to the receiving phase in the back-extraction step.

The results for *E*_tot,LA_ in Figure 2 show a similar trend for HCl, HNO_3_, H_2_SO_4_, and H_3_PO_4_. The *E*_tot,LA_ decreases from TOA:1-octanol to TOA:tmDES to TOA:limonene to TOPO:tmDES. The highest *E*_tot,LA_ is achieved with the mineral acid H_3_PO_4_ and the solvent phases TOA:1-octanol (29.8 ± 0.3%) and TOA:tmDES (23.1 ± 0.6%). This might lead to the conclusion that H_3_PO_4_ is the right choice for pH adjustment for LA extraction. However, the *E*_tot,mineral_ for H_3_PO_4_ in Figure 2d is 50.9 ± 0.1% with TOA:1-octanol and 36.5 ± 0.2% with TOA:tmDES indicating high mineral acid co-extraction. All solvent phases containing TOA as the reactive extractant show a significant co-extraction of HCl, HNO_3_, H_2_SO_4_, and H_3_PO_4_. Especially TOA:limonene exhibits high *E*_tot,mineral_ of 34.1 ± 1.1, 28.0 ± 0.2, 25.9 ± 0.3, and 17.8 ± 0.0 for HCl, HNO_3_, H_2_SO_4_, and H_3_PO_4_, respectively. In contrast, the solvent phase TOPO:tmDES shows a lower but more selective LA extraction when using HNO_3_, H_2_SO_4_, or H_3_PO_4_ for pH adjustment. An *E*_tot,LA_ of 5.37 ± 0.18, 5.94 ± 0.08, and 5.02 ± 0.01 with an *E*_tot,mineral_ of 0.0990 ± 0.0054, 0.0652 ± 0.0117, and 0.135 ± 0.011 is achieved for HNO_3_, H_2_SO_4_, and H_3_PO_4_.

## 3. Materials and Methods

This section summarizes the utilized materials and chemicals. Additionally, the employed analytical techniques and the protocols for conducting extraction and back-extraction experiments are outlined.

### 3.1. Materials and Chemicals

Table 6 lists all chemicals used in this study. The tmDES consisting of 60 mol% thymol and 40 mol% L-menthol was prepared as described in our previous work [6]. Model solutions were prepared with distilled water.

### 3.2. Analytical Methods

#### 3.2.1. High-Performance Liquid Chromatography

The LA concentrations in aqueous samples were analyzed using a high-performance liquid chromatography (HPLC) system (Dionex UltiMate 3000 from Thermo Fisher Scientific, Waltham, MA, USA) with a UV–Vis detector (operated at 210 nm) and a REZEX-ROA column (Rezex™ ROA-Organic Acid H+ 8%, LC Column 300 × 7.8 mm, Ea from Phenomenex). The mobile phase (flow rate 0.5 mL·min^−1^) was a 0.0025 M H_2_SO_4_ solution prepared with a 1 N H_2_SO_4_ solution (Carl Roth, Karlsruhe, Germany) and ultrapure water (CB1703, Adrona SIA, Riga, Latvia). The column oven temperature (Dionex STH 585 from Thermo Fisher Scientific, Waltham, Massachusetts, USA) was 40 °C. All HPLC samples were diluted with 0.0025 M H_2_SO_4_ solution, and DMSO was added as an internal standard.

#### 3.2.2. Ion Chromatography

Chlorine, nitrate, sulfate, and phosphate concentrations were measured in a Dionex Integrion ion chromatography system (IC) with a continuously regenerated anion trap column Dionex CR ATC 600 and a Dionex As-DV autosampler (Thermo Fisher Scientific, Waltham, MA, USA)). For eluent generation, potassium hydroxide from a Dionex EGC 500 KOH eluent generator cartridge and ultra-pure water were used. The system was equipped with a Dionex IonPac AG11-HC (2 × 50 mm) guard column, a Dionex IonPac AS11 HC (2 × 250 mm) analytical column, and a Thermo Scientific (Waltham, MA, USA) conductivity cell with a Dionex ASRS 300 2 mm electrolytically self-regenerating suppressor. Sample dilution was conducted with ultra-pure water. To increase the accuracy of the measurement, formate (TraceCERT^®^ 1000 mg·L^−1^ formate IC standard, Merck, Darmstadt, Germany) was added as an internal standard.

#### 3.2.3. pH Value, Density, and Water Content Measurement

The pH value of the aqueous samples was measured using a WTW SenTix^®^ 41 electrode with an integrated temperature sensor and a Knick (Berlin, Germany) Portavo^®^ 904(X) PH pH meter. The density of aqueous and organic phases was measured in an Anton Paar (Graz, Austria) DMA 45 density meter connected to a thermostat (±0.5 °C). The water content of organic phases was analyzed in an SI Analytics Titrator TitroLine^®^ 7500 KF with an Aquastar^®^ solvent for volumetric Karl Fischer titration (Supelco^®^, Merck, Darmstadt, Germany). Hydranal^®^-Titrant 5 (Honeywell Fluka, Charlotte, NC, USA) was used as a titrant.

### 3.3. Single-Stage Phase Equilibrium Measurements

Single-stage phase equilibrium measurements were performed in temperature-controlled separatory funnels connected to a thermostat (M3 MS, Lauda, Lauda-Königshofen, Germany) and mounted on a laboratory shaker (SM 25, Edmund Bühler, Bodelshausen, Germany). The phases were mixed for 60 min at 170 rpm, followed by 60 min of gravitational settling. All experiments were performed in duplicate to ensure reproducibility.

#### 3.3.1. Extraction

Three types of feed phases were used in the extraction experiments: a single-acid model solution containing one mineral acid, a multi-acid model solution comprising four mineral acids, and an LA model solution containing LA and one mineral acid. The organic solvent phases consisted of 0.2 mol·L^−1^ reactive extractant (TOA, TOPO, or TBP) in 1-octanol, tmDES, or limonene. The experiments were conducted with a volumetric phase ratio between the aqueous feed phase and the solvent phase of one (10 mL of each phase), and the temperature was set to 25 ± 0.5 °C. The evaluation of the extraction experiments was conducted using the mole-based distribution ratio *D*_extr,acid_ according to Equation (1).
*D*_extr,acid_ = (*c*_acid,E_ · *V*_E_)/(*c*_acid,R_ · *V*_R_)(1)

The extract phase volume *V*_E_ [L] and the raffinate phase volume *V*_R_ [L] were calculated using mass balances accounting for the volume change due to water transport from the aqueous feed phase into the extract phase, e.g., hydrogen-bonded to the extractant molecules or extracted acid molecules (see Appendix A). The solvent phases were assumed to have negligible solubility in the aqueous phase [6]. The acid concentration in the raffinate phase *c*_acid,R_ [mol·L^−1^] was measured using HPLC, and the acid concentration in the extract phase *c*_acid,E_ [mol·L^−1^] was determined using mass balances (see Appendix A) [6]. To evaluate the selectivity of the LA extraction over the extraction of the mineral acids, the selectivity *S*_extr,LA_, as defined by Equation (2), was calculated using the mole-based distribution ratio of LA and the mineral acid, denoted by *D*_extr,LA_ and *D*_extr,mineral_, respectively.
*S*_extr,LA_ = *D*_extr,LA_/*D*_extr,mineral_(2)

Moreover, the extractant loading *z*_acid_ and the total extractant loading *z*_tot_ were calculated according to Equations (3) and (4) using the moles of the respective acid in the extract phase *n*_acid,E_ [mol] and the moles of extractant in the extract phase *n*_extractant,E_ [mol].
*z*_acid_ = *n*_acid,E_/*n*_extractant,E_(3)
*z*_tot_ = (Σ*n*_acid,E_)/*n*_extractant,E_(4)

#### 3.3.2. Back-Extraction

The back-extraction experiments were conducted by mixing the extract phase with the antisolvent *p*-cymene. Distilled water was used as the receiving phase. To improve the recovery of the acids from the extract phase, the back-extraction was performed at an elevated temperature of 70 ± 0.5 °C. The volumetric extract phase-to-antisolvent phase ratio was three, and the volumetric extract phase-to-receiving phase ratio was two (5 mL extract phase, 15 mL antisolvent, and 10 mL receiving phase). The evaluation of the back-extraction experiments was conducted using the mole-based distribution ratio *D*_back,acid_ according to Equation (5).
*D*_back,acid_ = (*c*_acid,LR_ · *V*_LR_)/(*c*_acid,ASEX_ · *V*_ASEX_)(5)

The loaded receiving phase volume *V*_LR_ [L] and volume of the mixture of extract phase and antisolvent after the back-extraction *V*_ASEX_ [L] were determined by mass balances accounting for the water transport between the phases according to the same principle as for the extraction step. The acid concentration in the loaded receiving phase *c*_acid,LR_ [mol·L^−1^] was measured by HPLC, and the acid concentration in the extract phase-antisolvent mixture *c*_acid,ASEX_ [mol·L^−1^] was determined by mass balances. The selectivity of the LA back-extraction over the back-extraction of the mineral acid *S*_back,LA_ was calculated according to Equation (6), where *D*_back,LA_ denotes the mole-based distribution ratio of LA and *D*_back,mineral_ denotes the mole-based distribution ration of the respective mineral acid.
*S*_back,LA_ = *D*_back,LA_/*D*_back,mineral_(6)

To assess the efficiency of the extraction and back-extraction, the total efficiency *E*_tot_ [%] was calculated according to Equation (7) using the moles of acid in the loaded receiving phase and the feed phase, denoted by *n*_acid,LR_ [mol] and *n*_acid,F_ [mol], respectively.
*E*_tot,acid_ = *n*_acid,LR_/*n*_acid,F_(7)

## 4. Conclusions

In this study, we evaluated the co-extraction of hydrochloric, nitric, sulfuric, and phosphoric acid during reactive extraction of lactic acid from model solutions. The solvent phase consisted of a reactive extractant (tri-*n*-octylamine, trioctylphosphine oxide, or tributyl phosphate) diluted in a thymol-menthol-based deep eutectic solvent, limonene, or 1-octanol. Moreover, the co-back-extraction of the mineral acids was investigated for the back-extraction of lactic acid using a diluent swing with *p*-cymene as the antisolvent and water as the receiving phase. The tertiary amine tri-*n*-octylamine exhibited the highest total lactic acid efficiency (summarizing the extraction and back-extraction) of up to 29.8% but resulted in the highest mineral acid co-extraction of up to 50.9% in single-stage extractions. With trioctylphosphine oxide, a lower but more selective lactic acid extraction (5.94%) was achieved, with a lower mineral acid co-extraction (0.135%). The highest mineral acid co-extraction was observed for phosphoric acid and the lowest for nitric acid. In conclusion, the results show that the solvent phase composition and mineral acid influence the co-extraction and, hence, the final product purity. Moreover, the successful application of the natural deep eutectic solvent as a modifier enhances the sustainability of liquid-liquid extraction processes.

## Figures and Tables

**Figure 1 molecules-29-01722-f001:**
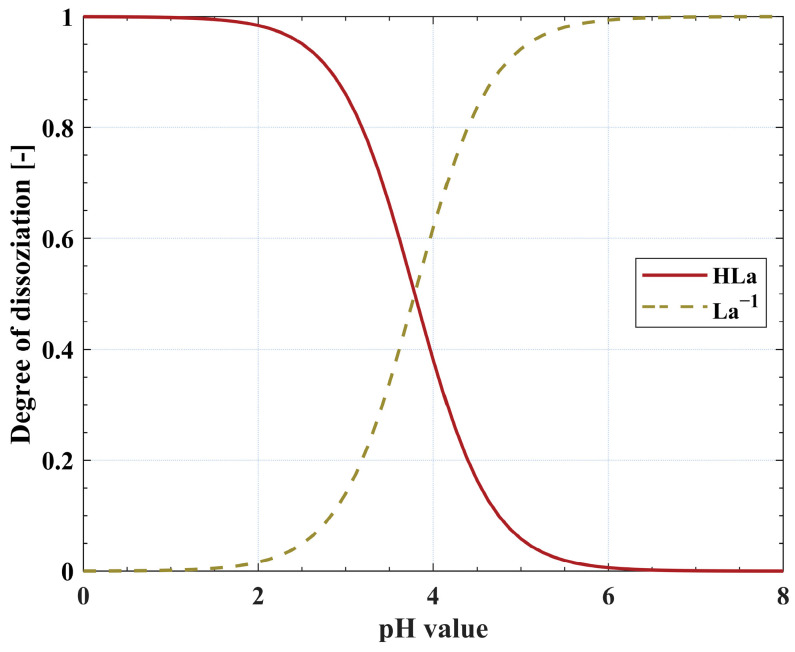
Dissociation diagram showing the equilibrium distribution between protonated (HLa) and deprotonated LA molecules (La^−1^) in dependence on the pH value. The p*K*_a_ value of LA is 3.79 [24].

**Figure 2 molecules-29-01722-f002:**
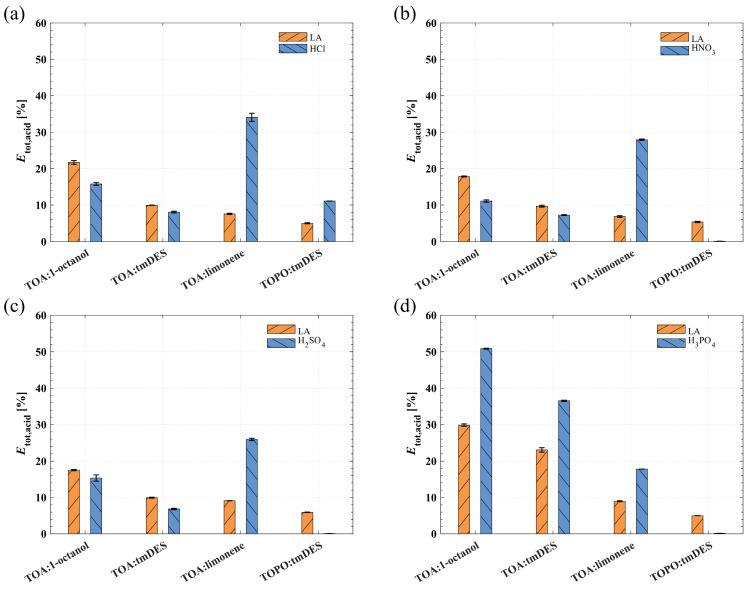
The total efficiency (*E*_tot,acid_) for LA and the mineral acids (**a**) HCl, (**b**) HNO_3_, (**c**) H_2_SO_4_, and (**d**) H_3_PO_4_. The extraction was performed at 25.0 ± 0.5 °C at a volumetric phase ratio of one. The back-extraction was examined at 70.0 ± 0.5 °C at a volumetric phase ratio of 1:2:3 (extract phase:receiving phase:antisolvent).

**Table 1 molecules-29-01722-t001:** The feed pH value (pH_F_), raffinate pH value (pH_R_), and mole-based distribution ratio (*D*_extr,mineral_) of HCl, HNO_3_, H_2_SO_4_, and H_3_PO_4_ for the physical extraction from single-acid model solutions using 1-octanol, tmDES, and limonene at 25 ± 0.5 °C. The volumetric feed-to-solvent phase ratio was one.

Acid	pH_F_	Solvent Phase	pH_R_	*D* _extr,mineral_
HCl	0.46 ± 0.01	1-Octanol	0.48 ± 0.03	0.0359 ± 0.0042
		tmDES	0.50 ± 0.01	0.0480 ± 0.0011
		Limonene	0.49 ± 0.00	0.0670 ± 0.0026
HNO_3_	0.46 ± 0.01	1-Octanol	0.52 ± 0.00	0.0533 ± 0.0072
		tmDES	0.50 ± 0.01	0.0384 ± 0.0006
		Limonene	0.48 ± 0.00	0.0375 ± 0.0031
H_2_SO_4_	0.47 ± 0.01	1-Octanol	0.49 ± 0.01	0.0385 ± 0.0020
		tmDES	0.48 ± 0.01	0.0649 ± 0.0024
		Limonene	0.45 ± 0.01	0.0877 ± 0.0032
H_3_PO_4_	1.15 ± 0.01	1-Octanol	1.19 ± 0.00	0.0743 ± 0.0017
		tmDES	1.18 ± 0.00	0.0806 ± 0.0009
		Limonene	1.16 ± 0.01	0.0826 ± 0.0075

**Table 2 molecules-29-01722-t002:** The raffinate pH value (pH_R_) and mole-based distribution ratio (*D*_extr,mineral_) for the reactive extraction of HCl, HNO_3_, H_2_SO_4_, and H_3_PO_4_ from the single-acid model solutions using TOA, TOPO, or TBP diluted in 1-octanol, tmDES, or limonene at 25 ± 0.5 °C. The volumetric feed-to-solvent phase ratio was one.

		**HCl**		**HNO_3_**	
**Solvent Phase**		**pH_R_**	** *D* _extr,mineral_ **	**pH_R_**	** *D* _extr,mineral_ **
TOA	1-Octanol	0.73 ± 0.00	22.5 ± 0.1	0.74 ± 0.00	22.3 ± 0.2
	tmDES	0.76 ± 0.01	1.27 ± 0.04	0.75 ± 0.00	1.27 ± 0.03
	Limonene	0.78 ± 0.01	1.18 ± 0.01	0.76 ± 0.00	1.23 ± 0.01
TOPO	1-Octanol	0.49 ± 0.00	0.0520 ± 0.0018	0.51 ± 0.00	0.0794 ± 0.0050
	tmDES	0.47 ± 0.00	0.0205 ± 0.0048	0.51 ± 0.00	0.0239 ± 0.0069
	Limonene	0.48 ± 0.00	0.0234 ± 0.0008	0.51 ± 0.00	0.0853 ± 0.0010
TBP	1-Octanol	0.47 ± 0.00	0.0100 ± 0.0018	0.50 ± 0.02	0.0794 ± 0.0114
	tmDES	0.49 ± 0.01	0.0746 ± 0.0029	0.49 ± 0.00	0.0539 ± 0.0003
	Limonene	0.49 ± 0.00	0.0570 ± 0.0011	0.49 ± 0.00	0.0436 ± 0.0046
		**H_2_SO_4_**		**H_3_PO_4_**	
**Solvent Phase**		**pH_R_**	** *D* _extr,mineral_ **	**pH_R_**	** *D* _extr,mineral_ **
TOA	1-Octanol	0.61 ± 0.01	17.3 ± 0.1	1.37 ± 0.00	16.1 ± 0.0
	tmDES	0.62 ± 0.00	0.776 ± 0.008	1.34 ± 0.02	0.402 ± 0.009
	Limonene	0.63 ± 0.00	0.686 ± 0.007	1.23 ± 0.00	0.386 ± 0.006
TOPO	1-Octanol	0.46 ± 0.01	0.0633 ± 0.0077	1.27 ± 0.00	0.124 ± 0.002
	tmDES	0.44 ± 0.00	0.0888 ± 0.0016	1.17 ± 0.01	0.114 ± 0.015
	Limonene	0.44 ± 0.00	0.0332 ± 0.0049	1.17 ± 0.00	0.121 ± 0.001
TBP	1-Octanol	0.44 ± 0.00	0.0460 ± 0.0065	1.14 ± 0.00	0.129 ± 0.015
	tmDES	0.45 ± 0.00	0.0737 ± 0.0030	1.13 ± 0.00	0.130 ± 0.011
	Limonene	0.46 ± 0.00	0.0825 ± 0.0043	1.17 ± 0.00	0.117 ± 0.001

**Table 3 molecules-29-01722-t003:** Raffinate pH value (pH_R_), total extractant loading (*z*_tot_), and mole-based distribution ratio (*D*_extr,mineral_) of HCl, HNO_3_, H_2_SO_4_, and H_3_PO_4_ for the reactive extraction from the multi-acid model solution using TOA, TOPO, or TBP diluted in 1-octanol, tmDES, or limonene at 25 ± 0.5 °C. The volumetric feed-to-solvent phase ratio was one.

				*D* _extr,mineral_			
Solvent Phase		pH_R_	*z* _tot_	HCl	HNO_3_	H_2_SO_4_	H_3_PO_4_
TOA	1-Octanol	0.92 ± 0.01	1.07 ± 0.02	2.06 ± 0.01	3.26 ± 0.03	1.09 ± 0.01	0.176 ± 0.002
	tmDES	0.87 ± 0.01	0.933 ± 0.002	1.82 ± 0.02	1.87 ± 0.02	0.573 ± 0.010	0.171 ± 0.002
	Limonene	0.92 ± 0.01	0.783 ± 0.008	1.19 ± 0.01	1.28 ± 0.01	0.392 ± 0.002	0.150 ± 0.019
TOPO	1-Octanol	0.65 ± 0.03	0.0780 ± 0.0134	0.0451 ± 0.0093	0.0551 ± 0.0067	0.0231 ± 0.0063	0.0383 ± 0.0060
	tmDES	0.60 ± 0.01	0.0840 ± 0.0096	0.0312 ± 0.0023	0.0419 ± 0.0063	0.0487 ± 0.0056	0.0417 ± 0.0055
	Limonene	0.65 ± 0.01	0.0596 ± 0.0152	0.0259 ± 0.0068	0.0209 ± 0.0066	0.0364 ± 0.0085	0.0361 ± 0.0090
TBP	1-Octanol	0.63 ± 0.01	0.0771 ± 0.0048	0.0497 ± 0.0019	0.0407 ± 0.0026	0.0312 ± 0.0041	0.0367 ± 0.0044
	tmDES	0.62 ± 0.01	0.0718 ± 0.0041	0.0450 ± 0.0031	0.0398 ± 0.0017	0.0339 ± 0.0005	0.0216 ± 0.0018
	Limonene	0.63 ± 0.01	0.0486 ± 0.0062	0.0118 ± 0.0037	0.0353 ± 0.0008	0.0348 ± 0.0022	0.0154 ± 0.0053

**Table 4 molecules-29-01722-t004:** The feed pH value (pH_F_), raffinate pH value (pH_R_), total extractant loading (z_tot_), mole-based distribution ratio of LA (*D*_extr,LA_), mole-based distribution ratio of mineral acids (*D*_extr,mineral_), and selectivity of LA extraction (*S*_extr,LA_) for the reactive extraction of LA from a model solution using TOA, TOPO, or TBP diluted in 1-octanol, tmDES, or limonene at 25 ± 0.5 °C. The model solutions contained LA and one mineral acid, and the volumetric feed-to-solvent phase ratio was one.

Mineral Acid	pH_F_	Solvent Phase	pH_R_	*z* _tot_	*D* _extr,LA_	*D* _extr,mineral_	*S* _extr,LA_
HCl	0.78 ± 0.01	TOA:1-octanol	1.94 ± 0.01	1.26 ± 0.01	0.429 ± 0.009	18.9 ± 0.2	0.0229 ± 0.0002
		TOA:tmDES	1.86 ± 0.03	1.09 ± 0.01	0.202 ± 0.006	7.42 ± 0.37	0.0295 ± 0.0003
		TOA:limonene	1.52 ± 0.00	0.979 ± 0.002	0.116 ± 0.003	5.75 ± 0.13	0.0201 ± 0.0002
		TOPO:tmDES	0.79 ± 0.01	0.0947 ± 0.0023	0.0782 ± 0.0033	0.0207 ± 0.0004	3.56 ± 0.03
HNO_3_	0.79 ± 0.01	TOA:1-octanol	2.05 ± 0.06	1.14 ± 0.01	0.276 ± 0.008	21.5 ± 0.3	0.0126 ± 0.0001
		TOA:tmDES	1.82 ± 0.03	1.07 ± 0.01	0.158 ± 0.007	13.0 ± 0.1	0.0125 ± 0.0001
		TOA:limonene	1.88 ± 0.04	1.01 ± 0.01	0.106 ± 0.006	8.97 ± 0.07	0.0113 ± 0.0001
		TOPO:tmDES	0.80 ± 0.00	0.098 ± 0.003	0.0828 ± 0.0023	0.0229 ± 0.0005	3.64 ± 0.04
H_2_SO_4_	0.80 ± 0.01	TOA:1-octanol	1.05 ± 0.00	0.806 ± 0.008	0.284 ± 0.007	1.56 ± 0.02	0.181 ± 0.002
		TOA:tmDES	1.01 ± 0.00	0.691 ± 0.031	0.153 ± 0.034	1.27 ± 0.01	0.146 ± 0.001
		TOA:limonene	1.10 ± 0.01	0.674 ± 0.011	0.145 ± 0.002	1.22 ± 0.04	0.122 ± 0.001
		TOPO:tmDES	0.74 ± 0.00	0.0939 ± 0.0036	0.0728 ± 0.0036	0.0261 ± 0.0002	2.63 ± 0.03
H_3_PO_4_	1.37 ± 0.01	TOA:1-octanol	1.79 ± 0.01	1.00 ± 0.01	0.877 ± 0.011	1.18 ± 0.04	0.776 ± 0.008
		TOA:tmDES	1.57 ± 0.00	0.908 ± 0.006	0.790 ± 0.010	0.836 ± 0.008	0.948 ± 0.009
		TOA:limonene	1.45 ± 0.00	0.296 ± 0.011	0.128 ± 0.012	0.222 ± 0.002	0.527 ± 0.005
		TOPO:tmDES	1.38 ± 0.00	0.0933 ± 0.0009	0.0740 ± 0.0003	0.0236 ± 0.0012	3.29 ± 0.03

**Table 5 molecules-29-01722-t005:** The pH value of the loaded receiving phase (pH_LR_), mole-based distribution ratio of LA (*D*_back,LA_), mole-based distribution ratio of the mineral acids (*D*_back,mineral_), and the selectivity of LA back-extraction (*S*_back,LA_) for back-extraction experiments. The extract phases were from experiments performed in Section 2.4, the back-extraction temperature was 70 ± 0.5 °C, and the volumetric phase ratio was 1:2:3 (extract phase:receiving phase:antisolvent).

Mineral Acid	Solvent Phase	pH_LR_	*D* _back,LA_	*D* _back,mineral_	*S* _back,LA_
HCl	TOA:1-octanol	1.90 ± 0.01	2.61 ± 0.15	0.200 ± 0.004	13.0 ± 0.5
	TOA:tmDES	2.19 ± 0.00	1.45 ± 0.01	0.101 ± 0.002	14.4 ± 0.4
	TOA:limonene	1.42 ± 0.00	2.72 ± 0.17	0.668 ± 0.026	4.07 ± 0.10
	TOPO:tmDES	3.15 ± 0.03	2.22 ± 0.15	0.0597 ± 0.0039	37.6 ± 4.9
HNO_3_	TOA:1-octanol	2.02 ± 0.01	4.71 ± 0.14	0.131 ± 0.003	35.9 ± 0.3
	TOA:tmDES	2.27 ± 0.02	2.48 ± 0.16	0.0846 ± 0.0010	29.3 ± 2.2
	TOA:limonene	1.69 ± 0.01	2.49 ± 0.20	0.451 ± 0.003	5.53 ± 0.48
	TOPO:tmDES	3.10 ± 0.03	2.37 ± 0.19	0.0463 ± 0.0019	51.5 ± 6.2
H_2_SO_4_	TOA:1-octanol	1.75 ± 0.03	3.79 ± 0.14	0.336 ± 0.018	11.3 ± 1.0
	TOA:tmDES	2.42 ± 0.02	3.07 ± 0.11	0.140 ± 0.003	22.0 ± 1.3
	TOA:limonene	1.29 ± 0.00	2.54 ± 0.02	0.896 ± 0.015	2.84 ± 0.07
	TOPO:tmDES	3.18 ± 0.00	2.65 ± 0.09	0.0263 ± 0.0034	102 ± 10
H_3_PO_4_	TOA:1-octanol	1.74 ± 0.00	1.77 ± 0.04	15.5 ± 0.4	0.114 ± 0.001
	TOA:tmDES	1.94 ± 0.00	1.10 ± 0.04	4.07 ± 0.07	0.269 ± 0.006
	TOA:limonene	2.11 ± 0.02	3.79 ± 0.18	53.1 ± 0.5	0.0715 ± 0.0026
	TOPO:tmDES	3.09 ± 0.01	2.68 ± 0.01	0.0624 ± 0.0040	43.2 ± 2.6

**Table 6 molecules-29-01722-t006:** Chemicals employed in this study.

Chemical	Shortcut	CAS	Purity	Supplier
Tri-*n*-octylamine	TOA	1116-76-3	98%	Sigma Aldrich, Darmstadt, Germany
Cyanex^®^ 921 (trioctylphosphine oxide)	TOPO	78-50-2	91%	Solvay, Hannover, Germany
Tributyl phosphate	TBP	126-73-8	97%	Sigma Aldrich, Darmstadt, Germany
1-Octanol		111-87-5	≥99%	Carl Roth, Karlsruhe, Germany
Thymol		89-83-8	≥99%	Carl Roth, Karlsruhe, Germany
L-Menthol		2216-51-5	≥99%	Sigma Aldrich, Darmstadt, Germany
(R)-(+)-Limonene		5989-27-5	95%	Carl Roth, Karlsruhe, Germany
*p*-Cymene		99-87-6	≥99%	Sigma Aldrich, Darmstadt, Germany
Lactic acid	LA	50-21-5	80%	Sigma Aldrich, Darmstadt, Germany
Hydrochloric acid	HCl	7647-01-0	37%	Carl Roth, Karlsruhe, Germany
Nitric acid	HNO_3_	7697-37-2	65%	Carl Roth, Karlsruhe, Germany
Sulfuric acid	H_2_SO_4_	7664-93-9	98%	Carl Roth, Karlsruhe, Germany
Phosphoric acid	H_3_PO_4_	7664-38-2	85%	Merck, Darmstadt, Germany
Dimethyl sulfoxide	DMSO	67-68-5	>99.91%	ThermoFischer Scientific, Waltham, MA, USA

## Data Availability

Data are contained within the article and Appendix A.

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
