# Peer review of "Mineral Acid Co-Extraction in Reactive Extraction of Lactic Acid Using a Thymol-Menthol Deep Eutectic Solvent as a Green Modifier"

_molecules, 2024, doi:10.3390/molecules29081722_

Round 1

Reviewer 1 Report

Comments and Suggestions for Authors

The presented manuscript, "Mineral acid co-extraction in reactive extraction of lactic acid using a thymol-menthol deep eutectic solvent as a green modifier", describes an interesting process for lactic acid co-extraction using greener solvents. The topic is relevant and offers valuable insights for readers. I recommend it for publication after minor revisions. Below are some comments to improve the manuscript:

Abstract: The abstract provides a clear overview of the study and the importance of natural deep eutectic as greener solvents. Also the methodology and results are correctly summarized.

Introduction: The introduction offers a comprehensive overview of the study's background, context, and objectives. It is well-structured. The introduction appropriately references several studies to support statements.

Material and Method section: This section offers a detailed description of the experimental setup.

Lines 176 – 179. Please considering include briefly or in the supplementary, information about mass balances considering the volume change. This information is crucial for phase equilibrium measurements.

Please, check the “%” in the column “purity” in table 1.

Please check title section 2.3.

Results and Discussion section:

Line 224. Please, be clear with the text, mineral acid extraction values are lower than 0.048.

Please consider use a comparison by medias for the values in tables 2 and 3, that information improves the understanding the treatments.

Conclusion section: The conclusion is well-written and summarizes the main points of the manuscript effectively.

Author Response

The presented manuscript, "Mineral acid co-extraction in reactive extraction of lactic acid using a thymol-menthol deep eutectic solvent as a green modifier", describes an interesting process for lactic acid co-extraction using greener solvents. The topic is relevant and offers valuable insights for readers. I recommend it for publication after minor revisions. Below are some comments to improve the manuscript:

Abstract: The abstract provides a clear overview of the study and the importance of natural deep eutectic as greener solvents. Also the methodology and results are correctly summarized.

Introduction: The introduction offers a comprehensive overview of the study's background, context, and objectives. It is well-structured. The introduction appropriately references several studies to support statements.

Material and Method section: This section offers a detailed description of the experimental setup.

Conclusion section: The conclusion is well-written and summarizes the main points of the manuscript effectively.

Dear Reviewer 1, thank you very much for carefully reading the manuscript and providing valid and very welcome feedback. We think your comments and suggestions have increased the quality of the manuscript.

Comment

Answer

Lines 176 – 179. Please considering include briefly or in the supplementary, information about mass balances considering the volume change. This information is crucial for phase equilibrium measurements.

Thank you for this remark! We have added the equations for the calculations in the supplementary materials.

In the manuscript we have added on page 5 line 184:

“(see Equation S1 to Equation S6 in the supplementary materials).”

As the back-extraction step on Page 6 line207 was evaluated using the same principle, we have added  “…according to the same principle as for the extraction step.”

Please, check the “%” in the column “purity” in table 1.

Thank you for this comment! The missing “%” signs were added.

Please check title section 2.3.

Thank you for detailed reading the manuscript! The title of section 2.3. was corrected.

Line 224. Please, be clear with the text, mineral acid extraction values are lower than 0.048.

Thank you very much for this comment! The correct value “0.0359 ± 0.0042” was inserted.

Please consider use a comparison by medias for the values in tables 2 and 3, that information improves the understanding the treatments.

Thank you for this comment! We are not sure what is exactly meant by “comparison by medias”. The experiments in Table 2 and Table 3 use the same feed phases (one mineral acid). With respect to the solvent phases, Table 2 uses solvent phases without reactive extractant (only physical extraction takes place), whereas in Table 3 a reactive extractant was added (reactive extraction takes place). As a result, the comparison of the feed phases or solvent phases in Table 2 and Table 3 is not possible.

Reviewer 2 Report

Comments and Suggestions for Authors

The manuscript entitled 'Mineral acid co-extraction in reactive extraction of lactic acid using a thymol-menthol deep eutectic solvent as a green modifier' describes the use of DES as a solvent for the extraction of organic and mineral acids and the subsequent back-extraction of lactic acid. The background and purpose of the work are adequately described and the methods are detailed. However, some revisions are needed to make the work publishable. One of the particular advantages of DES, namely its low toxicity, is not explained in the introduction. A paper on this aspect has recently been published and should be mentioned (10.1007/s11356-022-23362-5).

A critical aspect of the manuscript is the use of what the authors call "phosphorus-based extractants" or "reactive extractants". In reality, the chemical properties of tri-n-octylamine, trioctylphosphine oxide or tributyl phosphate are very different. In particular, tri-n-octylamine is the only compound that has a good basicity and it is therefore normal that it performs better. However, as it reacts with strong acids, it is not clear whether the system using this amine is recyclable. Some information on the recyclability of the system should be provided.

Instead, it is unclear how trioctyl phosphine oxide can be defined as a reactive extractant as it is not able to react with acids. The intended role of tributyl phosphate should also be clarified in the Introduction.

Finally, both in the abstract and in the conclusion, the authors state that 'the phosphorus-based extractants showed a lower but more selective lactic acid extraction'. In both cases, it is unclear to which systems these comparisons are being made. This should be clarified. In addition, further considerations should be made in the conclusions.

Comments on the Quality of English Language

Moderate editing of English language required

Author Response

The manuscript entitled 'Mineral acid co-extraction in reactive extraction of lactic acid using a thymol-menthol deep eutectic solvent as a green modifier' describes the use of DES as a solvent for the extraction of organic and mineral acids and the subsequent back-extraction of lactic acid. The background and purpose of the work are adequately described and the methods are detailed. However, some revisions are needed to make the work publishable.

Dear Reviewer 2, thank you very much for reading the present paper properly and giving very valid and welcome feedback. We have the feeling that by implementing your suggestions the quality of the paper was increased a lot.

Comment

Answer

One of the particular advantages of DES, namely its low toxicity, is not explained in the introduction. A paper on this aspect has recently been published and should be mentioned (10.1007/s11356-022-23362-5).

Thank you for this comment! We have added information about the toxicity of NADES on page 2 line 69 of the manuscript:

“As NADES consist of naturally derived compounds, they are considered biocompatible, environmentally friendly, and non-toxic [17]. Nevertheless, toxicological studies are still scarce and the available literature only investigates the toxicity of hydrophilic NADES. However, the available studies show that the toxicity of hydrophilic NADES depends on their constituents and concentration [18–22].”

However, available toxicity studies deal with hydrophilic NADES (e.g. choline chloride based NADES) and no studies are available for hydrophobic NADES.

A critical aspect of the manuscript is the use of what the authors call "phosphorus-based extractants" or "reactive extractants". In reality, the chemical properties of tri-n-octylamine, trioctylphosphine oxide or tributyl phosphate are very different. In particular, tri-n-octylamine is the only compound that has a good basicity and it is therefore normal that it performs better. However, as it reacts with strong acids, it is not clear whether the system using this amine is recyclable. Some information on the recyclability of the system should be provided.

Instead, it is unclear how trioctyl phosphine oxide can be defined as a reactive extractant as it is not able to react with acids. The intended role of tributyl phosphate should also be clarified in the Introduction.

Thank you for this remark! The better performance of TOA as compared to TOPO and TBP is well known, but we have chosen to include TOPO and TBP in our study as they are commonly applied in carboxylic acid extraction. We included the following part on page 3 line 103 of the manuscript:

“As reactive extractants, TOA, TOPO, and tributyl phosphate (TBP) were used, which are commonly applied for carboxylic acid extraction [32–34].”

We understand your point of view. However, one group of extractants defined in reactive extraction literature is termed “solvating agents”. This group includes the extractants trioctylphosphine oxide and tributyl phosphate.

Compare for example:

Reactive Extraction by H.-J. Bart (10.1007/978-3-662-04403-2)

Reactive Extraction and Critical Raw Materials: Industrial Recovery of Tungsten by Willersinn and Bart (10.1002/cite.201600079)

However, we have added the following sentences on page 2 line 58 of the manuscript to make it more clear:

“Reactive extractants are categorized according to their extraction mechanisms, such as anion exchangers (e.g. secondary or tertiary amines) or solvating agents (e.g. phosphine oxides or alkyl-substituted phosphates) [13].”

Concerning the recyclability, the high distribution coefficients for the back-extraction step indicate good recyclability of the solvent phases. We have added the following sentences on page 12 line 429:

“The Dback,mineral and Dback,LA indicate good back-extractability of the mineral acids and LA from the extract phases suggesting good solvent phase recyclability. However, a more detailed investigation of the recyclability of the solvent phase is required in future work.”

Finally, both in the abstract and in the conclusion, the authors state that 'the phosphorus-based extractants showed a lower but more selective lactic acid extraction'. In both cases, it is unclear to which systems these comparisons are being made. This should be clarified. In addition, further considerations should be made in the conclusions.

Thank you very much for this comment! We have made it more clear by explicitly naming the extractants in the abstract on page 1 line 18:

“Tri-n-octylamine showed the highest efficiency for lactic acid (up to 29.8 %) but also the highest co-extraction of mineral acids (up to 50.9 %). In contrast, trioctylphosphine oxide exhibited a lower but more selective lactic acid extraction (5.94 %) with low mineral acids co-extraction (0.135 %).”

And in the conclusion on page 13 line 462:

“The tertiary amine tri-n-octylamine exhibited the highest total lactic acid efficiency (summarizing the extraction and back-extraction) of up to 29.8 % but resulted in the highest mineral acid co-extraction of up to 50.9 % in single-stage extractions. With trioctylphosphine oxide, a lower but more selective lactic acid extraction (5.94 %) was achieved, with a lower mineral acids co-extraction (0.135 %).”

Moderate editing of English language required

The manuscript was checked and adapted accordingly.

Round 2

Reviewer 2 Report

Comments and Suggestions for Authors

I would like to thank the authors for accepting my suggestions, and the answers given are satisfactory. Therefore, I would like to recommend the work for publication.